# Comparison of Clinical Outcomes, Risks, and Costs for 20,910 Donor In Vitro Fertilization and 16,850 Donor Artificial Insemination Treatment Cycles: A Retrospective Analysis in China

**DOI:** 10.3390/jcm12030954

**Published:** 2023-01-26

**Authors:** Xue-Feng Luo, Hui-Lan Wu, Xi-Ren Ji, Yu-Lin Tang, Wen-Jun Zhou, Zeng-Hui Huang, Qian Liu, Li-Qing Fan, Chuan Huang, Wen-Bing Zhu

**Affiliations:** 1Reproductive and Genetic Hospital of CITIC-Xiangya, Changsha 410006, China; 2Institute of Reproductive and Stem Cell Engineering, Basic Medicine College, Central South University, Changsha 410000, China

**Keywords:** cost-effectiveness, artificial insemination, in vitro fertilization, sperm donor, human sperm bank

## Abstract

Purpose: To evaluate the effectiveness of donor in vitro fertilization (IVF-D) and donor artificial insemination (AI-D) in clinical outcomes, risks, and costs. Methods: This study analyzed the cycle changes and clinical outcomes in 20,910 IVF-D and 16,850 AI-D cycles between 2013 and 2021 in the Reproductive and Genetic Hospital of CITIC-Xiangya. A cost-effectiveness analysis was performed to evaluate the costs per couple and per live birth cycle in the two treatment groups. Results: IVF-D had higher pregnancy and live birth rates than AI-D (*p* < 0.001). The cumulative pregnancy and live birth rates for three AI-D cycles were 41.01% and 32.42%, respectively, higher than the rates for one or two AI-D cycles. The multiple birth and birth defect rate of AI-D was lower than that of IVF-D significantly. IVF-D mean cost per couple was higher than that of AI-D (CNY32,575 vs. CNY11,062, *p* < 0.001), with a mean cost difference of CNY21,513 (95% confidence interval, CNY20,517–22,508). The mean costs per live birth cycle for IVF-D and AI-D were CNY49,411 and CNY31,246, respectively. Conclusion: AI-D is more cost-effective and poses a lower risk for infertility couples than IVF-D, and patients should undergo three AI-D cycles to obtain the highest success rate.

## 1. Introduction

It was reported that the incidence of infertility in China has risen from 12 to 18% between 2007 and 2020 [1]. Infertility has become the third leading disease affecting human health. Many couples seek assisted reproductive technologies to conceive and ultimately give birth to healthy offspring. The ART include two treatment types, in vitro fertilization (IVF) and artificial insemination (AI). In IVF, fertilization occurs outside the body, and then the zygote is transferred to the uterus. In AI, spermatozoa is used for intrauterine insemination (IUI), intracervical insemination (ICI), and intravaginal insemination (IVI) for in vivo fertilization. Due to the difference in the fertilization process, IVF has a higher fertilization rate but is more invasive than AI.

Its high success rate endows IVF treatment with an overbearing predominance in ART among patients with infertility so that other treatments, such as IUI, are relegated. Guidelines on assisted reproduction suggest that women with unexplained infertility, age <40, should undergo three to four IUI cycles before considering IVF [2]. However, IVF is often the patients’ primary choice, forgoing AI because of its perceived poor success rate, uncontrolled multiple births, and high cost due to repeated failures over several cycles [3]. Some studies compared the effectiveness of IVF/intracytoplasmic sperm injection and IUI treatment cycles, showing that although IUI had a lower success rate than IVF, it was more cost-effective in delivering live births and had a lower risk of maternal complications [3,4,5,6]. Furthermore, a systematic review found no difference in the live birth rate between IVF and IUI, with a low risk of ovarian hyperstimulation syndrome (OHSS) for both [7].

Patients using donor sperm often choose IVF-D as their primary treatment, particularly those whose first AI-D treatment has failed. No comprehensive and direct comparison between IVF-D and AI-D treatments for success rates, risks, and costs were reported. We aimed to determine whether IVF-D was more effective than AI-D. We compared the success rates, risks, and costs of AI-D with IVF-D as practiced in the Reproductive and Genetic Hospital of CITIC-Xiangya (Hunan Province Human Sperm Bank) between 2013 and 2021.

## 2. Materials and Methods

### 2.1. Study Design and Population

This retrospective study analyzed the cycle changes of IVF-D and AI-D treatment from 2013 to 2021. However, we analyzed the clinical pregnancy outcomes only for 2014–2019 because of the delay in reporting them. AI-D included donor IUI, ICI, and IVI. Data were obtained from the Reproductive and Genetic Hospital of CITIC-Xiangya (Hunan Province Human Sperm Bank). Without reports with no feedback and no obstetric outcomes, we included 20,910 IVF-D cycles and 16,850 AI-D cycles in this study, more details was shown in Appendix A. The Ethics Committee of the Reproductive and Genetic Hospital of CITIC-Xiangya approved this study (LL-SC-2021-020).

### 2.2. Criteria for Screening Sperm Donors in China

The sperm donors used in this study underwent a multi-step selection process that included semen analysis, medical assessment, and blood tests. All donors gave informed consent for using their semen samples. Semen samples were collected by masturbation after 2–7 days of abstinence.

The guidelines for sperm donors are as follows [8]: (a) age, 22–45 years; (b) in good health based on physical and psychological examination by a qualified doctor and with no family history of genetic diseases; (c) fresh semen should liquefy in <60 min, sperm concentration ≥60 × 10^6^/mL, progressive sperm motility ≥60%, and normal sperm morphology of >4%; (d) post-thaw semen should have ≥40% motility, ≥12 × 10^6^ motile sperm per vial, and a survival rate of ≥60%; (e) underwent karyotype analysis and laboratory testing to exclude individuals at high risk of having genetic diseases and sexually transmitted infections, including HIV-1, HIV-2, hepatitis B and C, syphilis, gonorrhea, mycoplasma, chlamydia, cytomegalovirus, *Toxoplasma gondii*, rubella virus, and herpes simplex virus types 1 and 2. Donors complying with all the above requirements are recruited, and their samples are cryopreserved and maintained for at least six months before use to rescreen for HIV.

### 2.3. Criteria for Selecting of IVF-D and AI-D in China

According to the Notice of the Ministry of Health on the revision of technical specifications, basic standards and ethical principles related to human assisted reproductive technology and human sperm banks in China [9], infertility patients using donor sperm need to meet the criteria, as follows: (1) irreversible azoospermia, severe oligospermia, asthenospermia and teratospermia, (2) failure of vasectomy, (3) ejaculatory disorder, (4) husband and/or family have a high risk of transmitting a genetic disorder to the offspring, (5) blood incompatibility between mother and child cannot lead to a viable newborn (note: severe deficits in semen quality in couples who do not wish to undergo ICSI need to sign the informed consent). Patients of IVF-D including: (a) gamete transport disorder caused by various factors in the female such as oviduct obstruction and inflammation; (b) ovulatory obstacle; (c) endometriosis; (d) immunologic infertility; (e) unexplained infertility. The selection of AI-D need to meet the criteria: (a) unexplained infertility; (b) polycystic ovarian syndrome; (c) cervical factor; (d) mild/moderate endometriosis; (e) hypoovarianism. All patients with AI-D should be checked for fallopian tube patency to ensure that at least one fallopian tube is patent.

### 2.4. Clinical Outcomes

The clinical outcomes included clinical pregnancy, live birth, miscarriage, multiple birth, and birth defect rates. For IVF-D, we calculated the cumulative pregnancy and live birth rates by dividing the number of pregnancies or live births by the number of transferred fresh embryos.

### 2.5. Cost-Effectiveness

The per couple cost was obtained from the finance section of the Reproductive and Genetic Hospital of CITIC-Xiangya for 2019 and included medical costs such as surgical fees, inspection fees, sperm processing, embryo freezing and thawing, and medication fees. Non-medical expenses such as transportation and accommodation fees were excluded. We compared IVF-D and AI-D for the per couple, pregnancy cycle, and per live birth cycle costs to explore their cost-effectiveness.

### 2.6. Statistical Analysis

Statistical analysis was performed using IBM SPSS Statistics for Windows, Version 25.0 (IBM Corp., Armonk, NY, USA). Linear regression explored the change trends of AI-D and IVF-D cycles. The chi-squared test compared the groups for pregnancy, live birth, miscarriage, multiple birth, and birth defects rates. The relative risk (RR) differences and 95% confidence intervals (CI) were calculated. The difference between IVF-D and AI-D in the cost per couple was analyzed by independent-samples *t*-test. GraphPad Prism 8 measured IVF-D and AI-D cycle trends during 2013–2021. The significance level was set at *p* < 0.05.

## 3. Results

### 3.1. IVF-D and AI-D Cycles

The cycle changes of IVF-D and AI-D are shown in Figure 1. Significant downward trends were noted for AI-D (R^2^, 0.95; *p* < 0.001) and IVF-D (R^2^, 0.49; *p* = 0.035) cycles. We further explored the AI-D and IVF-D cycle changes at CITIC-Xiangya and other reproductive centers using the semen donation service from Hunan Province Human Sperm Bank (Appendix A). In the Reproductive and Genetic Hospital of CITIC-Xiangya, AI-D and IVF-D cycles declined significantly over time (R^2^, 0.98; *p* < 0.001 and 0.77; *p* = 0.002, respectively) while the annual ART cycles including AI and IVF/ICSI increased (Appendix A). A significant downward trend in AI-D cycles was noted in other reproductive centers (R^2^, 0.77; *p* = 0.001), while the IVF-D cycles remained relatively stable (R^2^, 0.12; *p* = 0.329; Appendix A).

### 3.2. Clinical Outcomes of IVF-D and AI-D

Compared to AI-D, IVF-D had a higher pregnancy rate [74.5 vs. 25.9%; *p* < 0.001; RR, 2.92 (95% CI, 2.84–2.99)], and live birth rate [62.4 vs. 20.3%; *p* < 0.001; RR, 3.08 (95% CI, 2.98–3.18)]. AI-D and IVF-D had similar total miscarriage rates. The detailed clinical outcomes of IVF-D and AI-D are summarized in Table 1. Furthermore, we analyzed the clinical outcomes of AI-D for various numbers of cycles during 2014–2019. The details are depicted in Table 2. We found that the cumulative pregnancy and live birth rates for three, four, and five AI-D cycles were significantly higher than for one or two AI-D cycles (*p* < 0.005). The cumulative pregnancy and live birth rates for three AI-D cycles were 41.0 and 32.4%, respectively. Similar pregnancy and live birth rates following AI-D were noted when more than three cycles were used. Interestingly, the miscarriage rate following AI-D did not change with the increase in cycle number.

### 3.3. Risks of IVF-D and AI-D

In this study, the risk index of IVF-D and AI-D was mainly multiple birth and birth defect rate. Compared to AI-D, IVF-D resulted in significantly higher multiple birth rate [24.7 vs. 1.9%; *p* < 0.001; RR, 13.0 (95% CI, 10.46–16.12)] and birth defect rate [1.33% vs. 0.69%; *p* = 0.002; RR, 1.94 (95% CI, 1.27–2.96)]. The details are shown in Table 1.

### 3.4. Cost-Effectiveness of IVF-D and AI-D

The mean cost per couple for IVF-D was higher than that for AI-D (CNY32,575 vs. CNY11,062, *p* < 0.001), with a mean cost difference of ¥ 21,513 (95% CI, 20,517–22,508). The respective mean costs of IVF-D and AI-D were CNY42,295 and CNY25,743 per pregnancy and CNY49,411 and CNY31,246 per live birth (Table 3).

## 4. Discussion

The Reproductive and Genetic Hospital of CITIC-Xiangya (Hunan Province Human Sperm Bank) is the first human sperm bank established in China. It has provided sperm donation services to over 80 reproductive centers in 23 provinces, municipalities, and autonomous regions across the country (Appendix A). This study found that the annual number of IVF-D cycles was larger than that of AI-D cycles between 2015 and 2021(Appendix A). A similar result was reported by Bahadur et al. [3] who discovered an overall 10.4-fold increase in IVF use compared to IUI between 2012 and 2016 in the UK. Zhou et al. [10] retrospectively analyzed the data on IUI and IVF cycles in Beijing, China, between 2013 and 2015. They found that the number of IVF cycles had increased while that of IUI cycles had decreased. This phenomenon could be explained by the higher clinical pregnancy rate following IVF, encouraging many patients to choose IVF as their primary ART treatment, especially if the first IUI cycle failed to achieve a pregnancy.

For patients, the choice of ART depends mainly on the clinical outcomes and cost-effectiveness of the various methods. This research found that the clinical pregnancy and the live birth rates following IVF-D were higher than following AI-D, with respective ratios of 2.92:1 and 3.08:1. The difference in the live birth rate was larger than the previously reported 2.35:1 [3]. However, the cumulative pregnancy rate calculation in this study included fresh and frozen embryo transfer cycles. Regardless of whether the calculation is based on fresh embryo transfer cycles or cumulative pregnancy rate, many studies indicated that the clinical outcomes of IVF were better than those of IUI [11,12,13]. For this reason, many patients tend to choose IVF as their first-line treatment even though it is more invasive and associated with a higher risk of multiple pregnancies than AI [14,15].

Although the pregnancy rate following IVF-D was higher than following AI-D in this study, some researchers suggested that IVF and IUI had similar clinical outcomes. A recent randomized controlled trial (RCT) comparing the pregnancy outcomes of IVF with a single embryo transfer and IUI with ovarian stimulation in couples diagnosed with unexplained or mild male subfertility found similar live birth and clinical pregnancy rates in both treatment groups [16]. Another RCT compared the effectiveness of IVF-single embryo transfer (SET), IVF-modified natural cycle (MNC) and IUI-controlled ovarian hyperstimulation (COH) [6]. They found that the live birth and ongoing pregnancy rates differed insignificantly between the treatment arms, and the time-to-pregnancy for IVF-SET or IVF-MNC was similar to that for IUI-COH. The cumulative clinical pregnancy rate for three AI-D cycles in our study was 41.0%, higher than for one AI-D cycle. Our cumulative live birth rate for three AI-D cycles (32.4%) was similar to the previously reported 33.3% [17]. The National Institute for Health and Care Excellence (NICE) guidelines recommend that women be offered 3–6 cycles if treated by donor insemination (www.nice.org.uk/guidance/cg15, accessed on 13 October 2022). In China, 3–4 IUI cycles are recommended for women with unexplained infertility aged <40 years, and IVF is considered if unsuccessful [2]. Interestingly, we found that the pregnancy and live birth rates did not increase significantly when patients were treated with more than three AI-D cycles. Several well-designed studies showed that three IUI cycles were as successful as one IVF cycle [4,18,19]. Furthermore, it was shown that the cost of a successful pregnancy for patients with three IUI cycles was lower than that of an IVF treatment [20]. Therefore, we suggest that patients with infertility treated by AI-D should undergo three cycles to obtain the highest success rate.

IVF is more invasive than IUI, causing a higher patient burden. According to our findings, IVF-D resulted in higher multiple birth and birth defect rates than AI-D, although the small difference in birth defect rates might not be clinically significant. These results are presumably because more than one embryo is normally transferred during IVF cycles [21]. A retrospective cohort study analyzed a clinical database of 4776 donor oocyte-recipient IVF cycles, finding that more than one blastocyst was transferred in 48.7% of the cycles (2323/4774), resulting in a multiple pregnancy rate of 42.0% [22]. Besides, OHSS remains one of the most common complications of IVF cycles [23]. A recent study analyzing data from 2012–2016 in the UK found 781 OHSS cases in the IVF group and none in the IUI group [3]. Another study [4] also found no OHSS cases in the IUI group, while the incidence rate in the IVF treatment group was 3.7%. Considering those reports, we suppose that AI-D might pose a lower risk for females with infertility, although this study did not report the OHSS incidence.

When evaluating treatment effectiveness, we need to focus not only on the clinical outcomes but also on the costs. The mean costs per couple (CNY32,575 vs. CNY11,062) and per live birth cycle (CNY49,411 and CNY31,246) for IVF-D were significantly higher than for AI-D. AI-D was more cost-effective than IVF-D. Similarly, many studies have shown that IUI was more cost-effective than IVF [3,5,6]. One study [4] evaluated the cost per live birth for one IVF cycle and three IUI cycles, showing that the cost for IVF was 1.3-times higher than that for IUI. Another study, integrating success, risks, and costs to deliver a live birth, showed that while saving on maternal and neonatal costs, IUI was better than IVF [24].

Interestingly, our study found that the total number of IVF-D and AI-D cycles has significantly declined over the years, while the total ART cycles have increased at the Reproductive and Genetic Hospital of CITIC-Xiangya (Figure 1). We think that there were two causes for the cycles’ decline. First, the number of human sperm banks in China has increased to the present 27 (http://www.nhc.gov.cn/, accessed on 14 October 2022), resulting in a decrease in the semen supply traffic. Second, the increase in the application of the testicular sperm aspiration and extraction techniques in patients with nonobstructive azoospermia could deliver testicular sperm for ART, avoiding the need for donor sperm.

The current study had some limitations. First, as a retrospective study, we could not compare the cost-effectiveness of three AI-D cycles and one IVF-D cycle. Second, the data on birth defects were provided by reproductive centers, following the dictation of the donor semen recipient, possibly affecting the results. Finally, our findings are not based on the general male population; sperm donors were selected for their better sperm quality, and further studies of this type are needed. On the other hand, we did not calculate the cost of fresh and frozen embryos for IVF-D, because the cost of frozen embryo cycles was only a small part of the total cost of IVF-D; it did not affect the conclusion of the study. Furthermore, due to the principle of double blindness between sperm donors and recipients in China, we cannot collect more information about patients.

To our knowledge, this was the first study to compare the clinical outcomes and cost-effectiveness of IVF-D and AI-D. Furthermore, our retrospective analysis was based on the feedback data of over 80 reproductive centers across 23 provinces in China, which make our findings highly reliable.

## 5. Conclusions

In conclusion, although the pregnancy rate following IVF-D was higher than that following AI-D, AI-D had a lower multiple birth rate and lower costs per couple and live birth. Therefore, AI-D is more cost-effective and poses a lower risk for couples with infertility than IVF-D. Patients with infertility should undergo three AI-D cycles to obtain the highest success rate.

## Figures and Tables

**Figure 1 jcm-12-00954-f001:**
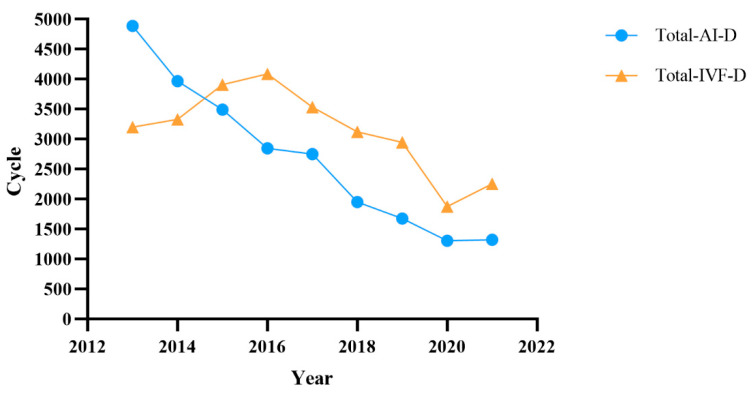
Total cycle changes of IVF-D and AI-D treatments.

**Table 1 jcm-12-00954-t001:** Clinical outcomes of IVF-D and AI-D during 2014–2019.

	IVF-D	AI-D	RR(IVF-D vs. AI-D)	95% CI	*p* Value
Pregnancy rate	74.45 (13683/18134)	25.89 (4362/16850)	2.92	2.84–2.99	0.000
Live birth rate	62.40 (11316/18134)	20.28 (3417/16850)	3.08	2.98–3.18	0.000
Miscarriage rate	13.56 (1855/13683)	14.65 (639/4362)	0.93	0.85–1.01	0.069
Multiple birth rate	24.74 (3385/13683)	1.90 (83/4362)	13.00	10.46–16.12	0.000
birth defect rate	1.33 (196/14724)	0.69 (24/3501)	1.94	1.27–2.96	0.002

**Table 2 jcm-12-00954-t002:** Clinical outcomes of AI-D cumulative cycle during 2014–2019.

Accumulation Cycle	Pregnancy Rate	Miscarriage Rate	Live Birth Rate
one	25.95 (2186/8424) b	15.65 (342/2186)	20.56 (1732/8424) b
two	36.64 (1878/5126) a	15.55 (292/1878)	28.79 (1476/5126) a
three	41.01 (2102/5126) ab	15.70 (330/2102)	32.42 (1622/5126) ab
four	42.29 (2168/5126) ab	15.54 (337/2168)	33.53 (1719/5126) ab
five	42.65 (2186/5126) ab	15.65 (342/2186)	33.79 (1732/5126) ab

a *p* < 0.005 when compared with cycle one. b *p* < 0.005 when compared with cycle two.

**Table 3 jcm-12-00954-t003:** Cost-effectiveness of IVF-D and AI-D.

Group	Cost/per Couple	Cost/per Pregnancy Cycle	Cost/per Live Birth Cycle
IVF-D	¥ 32,575 (*n* = 1127)	¥ 42,295 (*n* = 868)	¥ 49,411 (*n* = 743)
AI-D	¥ 11,062 (*n* = 370) *	¥ 25,743 (*n* = 159)	¥ 31,246 (*n* = 131)

* compared with IVF-D, *p* < 0.001.

## Data Availability

The data associated with the paper are not publicly available but are available from the corresponding author upon reasonable request.

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
