# Peer review of "Comparison of Clinical Outcomes, Risks, and Costs for 20,910 Donor In Vitro Fertilization and 16,850 Donor Artificial Insemination Treatment Cycles: A Retrospective Analysis in China"

_jcm, 2023, doi:10.3390/jcm12030954_

Round 1

Reviewer 1 Report

 This is a very important study on a very interesting clinical question and  well design. However, it’s very unclear how the focus of the conclusion is condusing. The pregnancy and  livebirth rate is around 3 times higher in the IVF-D and it doesn’t look it’s important. Additionally, the birth defect rate difference doesn’t look clinically significant.

Thus, this paper is not ready for publication as long as the discussion is not changing significantly and presenting the advantages of each method: IVF-D and AI-D and the conclusions as well.

Author Response

Response:

Thanks for the comments of the reviewer. In this study, the effectiveness of IVF-D and AI-D in clinical outcomes, risks and costs were evaluated, we found that although the pregnancy rate following IVF-D was higher than that following AI-D, AI-D had a lower multiple birth rate and lower costs per couple and live birth. Therefore, AI-D is more cost-effective and poses a lower risk for couples with infertility than IVF-D. In this study, the pregnancy and livebirth rate is around 3 times higher in the IVF-D, we depict it in the second paragraph of discussion. However, some researchers suggested that IVF and IUI had similar clinical outcomes. We also calculate the cumulative clinical pregnancy rate for three AI-D cycles, which is higher to 41.0%, but we don’t compare the clinical outcomes of three AI-D cycles and one IVF-D cycle, because we calculated the cumulative pregnancy and live birth rates for IVF-D, we depict it in the limitation. As for birth defect rate, according to our findings, IVF-D resulted in higher multiple birth and birth defect rates than AI-D, although the small difference in birth defect rates might not be clinically significant. We depict it in the fourth paragraph in discussion. From this result, we demonstrate that IVF is more invasive than IUI, causing a higher patient burden.

We have discussed the differences in clinical outcomes between IVF-D and AI-D as well as differences in cost of per couple, and drawn a conclusion: although the pregnancy rate following IVF-D was higher than that following AI-D, AI-D had a lower multiple birth rate and lower costs per couple and live birth. Therefore, AI-D is more cost-effective and poses a lower risk for couples with infertility than IVF-D. Patients with infertility should undergo three AI-D cycles to obtain the highest success rate.

Reviewer 2 Report

This is a retrospective study comparing the use of donor sperm by either artificial insemination or by IVF in two different ways, both the clinical effectiveness and the cost effectiveness.  The authors give a good explanation in the introduction why this type of study is needed, and I would like to commend them in this effort.  I do have a few comments that will hopefully improve the final manuscript:

1. I see that in table 2 you do report cumulative success rates for AI-D cycles in couples that did multiple cycles.  Aside from this analysis, it is difficult to see how you handled the possibility of multiple cycles per couple.  It looks to me like you are using only first cycle success rates to report your findings at the beginning of the paragraph in section 3.2, is that correct?  It would be good to state this more clearly.  And are there any couples that did multiple cycles of IVF?  Are these two groups mutually exclusive, meaning did you check to make sure there are no couples in both groups?  If there are, how did you account for this statistically?

2. Thank you for listing your inclusion and exclusion criteria.  Can you provide a figure showing how your narrowed down to the population you included?  How many couples were excluded from analysis and for what reasons?

3. Section 2.3 is a little unclear to me, as I am not as familiar with requirements in China.  Are these criteria showing who is allowed to proceed with treatment using donor sperm in China?

4. Do you have information to report on the number of embryos transferred in IVF cycles on average?

Author Response

Comment 1: I see that in table 2 you do report cumulative success rates for AI-D cycles in couples that did multiple cycles.  Aside from this analysis, it is difficult to see how you handled the possibility of multiple cycles per couple.  It looks to me like you are using only first cycle success rates to report your findings at the beginning of the paragraph in section 3.2, is that correct?  It would be good to state this more clearly.  And are there any couples that did multiple cycles of IVF?  Are these two groups mutually exclusive, meaning did you check to make sure there are no couples in both groups?  If there are, how did you account for this statistically?

Response:

Thanks for the comments of the reviewer. In this study, the beginning of the paragraph in section 3.2 is the comparison of clinical outcomes of IVF-D and AI-D during 2014-2019. The measurement of AI-D refers to the clinical outcomes of a per AI-D cycle, not just the first cycle. The reason why we calculate the clinical outcomes of AI-D cumulative cycle is showed in the third paragraph of the discussion. According to The National Institute for Health and Care Excellence (NICE) guidelines recommendation, women should be offered 3–6 cycles if treated by donor insemination. In China, 3–4 IUI cycles are recommended for women with unexplained infertility aged <40 years, and IVF is considered if unsuccessful. In this study, we found that the pregnancy and live birth rates did not increase significantly when patients were treated with more than three AI-D cycles. On the other hand, several well-designed studies showed that three IUI cycles were as successful as one IVF cycle. However, in this study, we calculate the cumulative pregnancy and live birth rates of IVF-D because the date is feedback from other reproduction center, so we don’t proceed the successful comparison of three AI-D cycles and one IVF-D cycle. But we suggest that patients with infertility treated by AI-D should undergo three cycles to obtain the highest success rate. About multiple cycles of IVF, the multiple cycles of IVF will be treat as multiple data record, so we think whether or not any couples had multiple cycles of IVF, this will not affect the conclusions of our study. And the couples in both group also be treat as multiple data record in two group.

Comment 2: Thank you for listing your inclusion and exclusion criteria.  Can you provide a figure showing how your narrowed down to the population you included?  How many couples were excluded from analysis and for what reasons?

Response:

Thanks for the comments of the reviewer. According to the inclusion and exclusion criteria, we have depicted a figure showing how many couples are excluded from analysis in this study, more details are shown in Supplementary Figures S3.

Supplementary Figures S3 cycles included in the study

Comment 3: Section 2.3 is a little unclear to me, as I am not as familiar with requirements in China.  Are these criteria showing who is allowed to proceed with treatment using donor sperm in China?

Response:

Thanks for the comments of the reviewer. According to the Notice of the Ministry of Health on the revision of technical specifications, basic standards and ethical principles related to human assisted reproductive technology and human sperm banks in China, infertility patients using donor sperm need to meet the criteria, as follows: (1) Irreversible azoospermia, severe oligospermia, asthenospermia and teratospermia (2) Failure of vasectomy (3) Ejaculatory disorder (4) The husband and/or family have a high risk of transmitting a genetic disorder to the offspring (5) Blood incompatibility between mother and child cannot lead to a viable newborn (note: severe deficits in semen quality in couples who do not wish to undergo ICSI need to sign the informed consent). We have described the criteria into section 2.3.

Comment 4: Do you have information to report on the number of embryos transferred in IVF cycles on average?

Response:

Thanks for the comments of the reviewer. I am sorry I don’t have the information of the number of embryos transferred in IVF cycles on average. In China, due to the principle of double blindness between sperm donors and recipients, we cannot collect more information.

Reviewer 3 Report

The study characterises efficiency od donor artificial in vitro fertilization with donor artificial insemination.

The results are very important for clinicals.

however much more information are needed on patients and clinics involved in the study.

I suggested more information on patents that were involved in the study and clinics that were share their data.It is a lot of information that Authors should add to improve their ms.

Author Response

Response:

Thanks for the comments of the reviewer. More information on patients can give us more information about the effectiveness of IVF-D and AI-D in clinical outcomes, risks and costs, such as the causes and years of infertility, infertility treatments. However, due to the principle of double blindness between sperm donors and recipients in China, we cannot collect more information about patients. Therefore, we have depicted it in limitation in our study.

Round 2

Reviewer 1 Report

Accepted for publication 

Author Response

Dear Reviewers,

Thank you for your comments regarding our manuscript entitled “Comparison of clinical outcomes, risks, and costs for 20,910 IVF-D and 16,850 AI-D treatment cycles: a retrospective analysis in China” (manuscript ID: jcm-2136009). Your comments are constructive and offered great insights to our future research. We have reviewed the comments very carefully and made revision, which were marked red in the revised manuscript. The main revision and the point-by-point response to the reviewer’ s comments are as followed.

Reviewer #1:

I only suggest better clarify the title, avoiding the use of acronyms (IVF-D and AI-D) to improve the readers' comprehension.

Response:

Thanks for the comments of the reviewer. We have changed the title of this study as “Comparison of clinical outcomes, risks, and costs for 20,910 donor in vitro fertilization and 16,850 donor artificial insemination treatment cycles: a retrospective analysis in China”, the acronyms IVF-D and AI-D were changed to donor in vitro fertilization and donor artificial insemination.

We appreciate your comments very much. We have tried our best to better the manuscript, and these changes will not change the main content and framework of the paper. We sincerely hope for your approval.

Once again, we sincerely thank you and your valuable comments.
